# Impaired dynein function preserves spinal interneuron survival and positioning in an ALS-like mouse model

Eleni Christoforidou[1], Jordan S. Rowe[1], Fabio A. Simoes[1], Raphaelle Cassel[2], Luc Dupuis[2], Peter Nigel Leigh[3], Majid Hafezparast[1]*

1 Department of Neuroscience, School of Life Sciences, University of Sussex, Brighton, United Kingdom, 2 University of Strasbourg, INSERM, Strasbourg, France, 3 Department of Neuroscience, Brighton & Sussex Medical School, University of Sussex, United Kingdom

* M.Hafezparast@sussex.ac.uk

## Abstract

Impaired cytoplasmic dynein function has been implicated in amyotrophic lateral sclerosis (ALS) pathogenesis, yet the contributions of spinal interneurons to disease phenotypes remain unclear. We tested the hypothesis that hypomorphic dynein function in cholinergic neurons disrupts the development, survival, or positioning of inhibitory interneuron populations in the lumbar spinal cord. Using ChAT-Cre recombination, we generated four mouse genotypes with graded reductions in dynein activity in ChAT⁺ cells: $Dync1h1^{+/+}$ (wildtype), $Dync1h1^{-/+}$ (hemizygous wildtype), $Dync1h1^{+/Loa}$ (heterozygous Loa mutation), and $Dync1h1^{-/Loa}$ (hemizygous Loa). At 52 weeks of age, lumbar spinal cords (L3–L6) were harvested, cryosectioned, and immunostained for ChAT, GAD-67, Parvalbumin, and Calbindin. Cell counts were performed on confocal images from eight sections per mouse (N = 3 male mice/genotype), and radial distances from the central canal were normalised to gray matter width. Angular distributions were analysed *via* circular statistics. There were no significant genotype-dependent differences in the numbers of ChAT⁺, GAD-67⁺, Parvalbumin⁺, or Calbindin⁺ cells, nor in ChAT⁺ subpopulations (motor neurons versus interneurons) or double-positive interneuron subsets (e.g., ChAT⁺–GAD-67⁺, Parvalbumin⁺–GAD-67⁺, Parvalbumin⁺–Calbindin⁺). Radial positioning relative to the central canal was similarly preserved across all markers and genotypes. Circular-median tests revealed statistically significant shifts in mean angle for ChAT⁺, GAD-67⁺, and certain double-positive cells, but these amounted to only 5–10° displacements, translating to lateral shifts of ~10–20 µm, well within single laminar bands, and are unlikely to impact circuit connectivity. Despite substantial motor deficits and hallmark TDP-43 pathology previously seen in these models, impaired dynein function does not precipitate interneuron loss or gross migratory defects in the lumbar spinal cord. Instead, our findings suggest that the primary contributions of dynein to ALS-like phenotypes likely arise

**Data availability statement:** All relevant data are within the manuscript and its Supporting Information files.

**Funding:** This work was funded by the Motor Neurone Disease Association grants Christoforidou/Nov24/2494-793 (to EC, MH, and PNL) and Dupuis/Apr16/852-791(to LD and MH). The funders of this project had no role in the study design, data collection and analysis, decision to publish, or preparation of the manuscript.

**Competing interests:** The authors have declared that no competing interests exist.

from functional disruptions in axonal transport, synaptic maintenance, and neuronal physiology rather than from structural alterations or loss of interneuron populations.

## Introduction

Amyotrophic Lateral Sclerosis (ALS) is a fatal neurodegenerative disorder characterised by the progressive loss of motor neurons, leading to muscle weakness, paralysis, and eventual death [1]. Despite extensive research, the underlying mechanisms remain poorly understood, and therapeutic options are limited. Interneurons have historically been neglected in ALS research partly because they are less visible to pathologists using conventional methods [2]. Traditional pathological examinations often miss these cells due to their small size and less prominent morphological characteristics compared to motor neurons. However, interneurons play a crucial role in modulating motor neuron activity and overall neural circuitry [3].

Pathological changes in interneurons have been documented in ALS [2], but the importance of these changes has not yet been established. Specifically, interneurons might experience changes in their excitability, connectivity, or neurotransmitter release in response to ALS pathology [4–6]. Moreover, muscle twitches (fasciculations) observed in ALS have traditionally been attributed to cell-autonomous defects in motor neurons [7]. However, recent studies suggest that these muscle twitches may also be linked to abnormal interneuron function [5,8].

Previous research showed changes in the density of some interneuron subtypes in the cortex of people with ALS [9]. Furthermore, a mouse model carrying a point mutation (M323K) within the endogenous *Tardbp* gene which codes for the TDP-43 protein shows progressive cortical interneuron degeneration [10]. Although mutations in *TARDBP* are causative for only ~4% of familial and ~1% of sporadic cases of ALS, TDP-43 pathology is present in ~97% of all ALS cases [11,12]. Additionally, findings from Project MinE [13], containing whole genome sequencing (WGS) data from a large number of ALS cases (6,538) and controls (2,415), have identified ALS patients with rare variants in genes not typically associated with ALS, suggesting these may contribute to disease susceptibility. One such gene is dynein, cytoplasmic heavy chain 1 (*DYNC1H1*) which encodes the heavy chain subunit of cytoplasmic dynein 1 (henceforth 'dynein'), a motor protein essential for neuronal cell motility and morphology, and the primary mediator of retrograde transport. Dynein has not consistently appeared in large-scale ALS genetic studies, but rare variants have been reported in motor neuron diseases, including certain ALS cases [14,15] and lower extremity–predominant spinal muscular atrophy (LED-SMA) [16,17]. While these variants are uncommon, they point to dynein dysfunction as a plausible contributor to disease mechanisms. As ALS is primarily sporadic, likely involving a combination of subthreshold genetic variants and environmental or lifestyle factors, gene variants in components of dynein may predispose individuals to ALS by perturbing interneuron networks within the motor neuron pools in the spinal cord. One possible cause of such perturbation is impaired dynein-mediated neuronal migration during neurodevelopment, which is a novel hypothesis in understanding ALS pathology. However, prior

research has shown that mutations in dynein or its accessory proteins can lead to cortical neuron migration defects and thus brain malformations [18].

The F580Y Legs at odd angles (*Loa*) mutation in the mouse *Dync1h1* disrupts dynein complex integrity and leads to defective cortical neuron migration, arborisation of neurons, and impaired retrograde axonal transport [19–24]. Importantly, our recent study has shown that the *Loa* mutation in mice can recapitulate cellular hallmarks of ALS by promoting aggregation of TDP-43 and p62 [25]. In another of our recent studies [26], we have shown that the combination of defective dynein with a *TARDBP* transgene (encoding TDP-43$^{M337V}$) in mice triggered TDP-43 pathology, partially recapitulating human ALS. These results suggest a complex interplay between protein aggregation and impaired dynein function, and underscore the need to further investigate the role of dynein in ALS.

To study the contribution of impaired dynein function to the onset of ALS, we previously generated mice harbouring a spectrum of the wildtype or *Loa* alleles of *Dync1h1* in cholinergic motor neurons using ChAT-Cre expressing mice. The ChAT-Cre line expresses Cre recombinase under the control of the endogenous choline acetyltransferase (*Chat*) promoter, with the inclusion of an internal ribosome entry site (IRES) allowing for the efficient expression of Cre in conjunction with the *ChAT* gene. The genotypes generated include: (1) *Dync1h1$^{+/+}$* (wildtype), (2) *Dync1h1$^{-/+}$* (hemizygous wildtype; one allele knocked out in ChAT$^+$ cells), (3) *Dync1h1$^{+/Loa}$* (heterozygous mutant; one allele carries the *Loa* mutation), and (4) *Dync1h1$^{-/Loa}$* (hemizygous mutant; one *Loa* mutant allele, one allele knocked out in ChAT$^+$ cells). This order was hypothesised to reflect a dose-dependent reduction in dynein function that leads to progressively worse ALS-like phenotypes.

In our previous characterisation of these mouse strains [25], both *Dync1h1$^{+/Loa}$* and *Dync1h1$^{-/Loa}$* mice displayed impairments in motor performance relative to wildtypes, including reduced grip strength detectable from approximately 4–8 weeks of age. These deficits persisted throughout adulthood and were accompanied by reduced body weight from around 8 weeks onwards. The more severe *Dync1h1$^{-/Loa}$* genotype exhibited an earlier and steeper decline in grip strength and a progressively greater reduction in body weight compared with *Dync1h1$^{+/Loa}$* mice, with the divergence between the two mutants becoming increasingly apparent with age and reaching statistical significance by ~40–52 weeks. In addition to these behavioural phenotypes, structural abnormalities at neuromuscular junctions (NMJs), including endplate fragmentation and reduced innervation, were already detectable at 8 weeks and persisted into later adulthood. Despite this, there was no loss of spinal motor neurons in our mouse models. It is possible that defects in the interneuron pools of the spinal cord may exacerbate the impact of aberrant NMJs, thereby contributing to the impaired motor function phenotype observed in these mice. This suggests that any aberrant interneuron migration could play a critical role in the overall motor deficits in these mice, highlighting the need for further investigation into the interplay between motor neurons and interneurons in this context.

Therefore, we hypothesised that disruption of dynein function may interfere with the development and/or survival of inhibitory interneurons in the spinal cord. To test this hypothesis, we explored whether there are changes in interneuron density and distribution in the spinal cord of our mouse models that exhibit other ALS-like pathology. Based on our previously observed phenotypic progression, we selected 52 weeks of age as a late adult timepoint at which motor deficits and associated neuromuscular pathology are well established. It is important to note that in these models the *Loa* mutation in *Dync1h1* is present in all cells, whereas the additional reduction in dynein function occurs selectively in cholinergic neurons through Cre-mediated deletion of the floxed *Dync1h1* allele. The inhibitory interneuron populations analysed in this study represent broader inhibitory classes that are not necessarily part of the ChAT lineage. Consequently, while these interneurons may carry the *Loa* mutation, they do not experience the same cholinergic-specific reduction in dynein function. As a result, the present study primarily examines whether dynein dysfunction in cholinergic neurons leads to indirect or circuit-level effects on the survival or spatial organisation of inhibitory interneuron populations within the lumbar spinal cord. By focusing on interneurons, we aimed to shed light on their potential contribution to ALS pathology, which has been underexplored, but which may be important in pathogenic mechanisms such as excitotoxicity [27–29].

## Materials and methods

### Animals

Procedures involving animals were approved by the University of Sussex Animal Welfare and Ethics Review Board (AWERB) and were conducted under a project licence issued by the UK Home Office in accordance with the UK Animals (Scientific Procedures) Act 1986. All experiments were performed in accordance with relevant institutional and national guidelines for the care and use of laboratory animals. Mice were on a C57Bl/6J background and were housed under a 12-hour light-dark cycle (lights on at 7 am) with free access to food and water and with up to five littermates. To generate mice with one *Dync1h1* allele knocked out only in cholinergic (ChAT$^+$) cells, a 3-step breeding program using Cre-Lox recombination was carried out as we previously described in [25]. This generated either wildtypes (*Dync1h1$^{+/+}$*), heterozygous mutants (*Dync1h1$^{+/Loa}$*), hemizygous wildtypes (*Dync1h1$^{-/+}$*) where one *Dync1h1* allele was absent only in ChAT$^+$ cells, or hemizygous mutants (*Dync1h1$^{-/Loa}$*) where again one *Dync1h1* allele was absent only in ChAT$^+$ cells and the Loa mutation was present in all cells. The genotyping procedure and the confirmation of successful recombination have also been previously described [25]. All procedures were conducted in accordance with the principles of the 3Rs. Animals were monitored regularly for health and welfare throughout the study by trained personnel in accordance with UK Home Office guidelines. Humane endpoints were predefined, and animals showing signs of pain, distress, or significant deterioration in condition (e.g., weight loss, impaired mobility, or abnormal behaviour) would have been humanely euthanised by perfusion fixation under terminal anaesthesia. No unexpected adverse events occurred during the course of the study.

### Perfusion and tissue collection

Only males were used in this study. This was done to maintain consistency with our previous characterisation of this mouse model, in which sex-dependent differences in body weight, grip strength and neuromuscular junction phenotypes were observed, with males showing a more pronounced and consistent phenotype. At 52 weeks of age, male animals were anaesthetised with an overdose of pentobarbital and transcardially perfused using a peristaltic pump with oxygenated artificial cerebrospinal fluid (125 mM NaCl, 2.5 mM KCl, 2.5 mM CaCl$_2$, 2 mM MgCl$_2$.6H$_2$O, 26 mM NaHCO$_3$, 1.25 mM NaH$_2$PO$_4$.H$_2$O, 25 mM C$_6$H$_{12}$O$_6$). The vertebral column was dissected and cut cross-sectionally between vertebrae T9 and T10 and between vertebrae L6 and S1. The lumbar portion of the vertebral column (T10-L6) was drop-fixed in 4% formaldehyde for 24 hours, followed by removal of the spinal cord from the vertebrae which was then stored in 30% sucrose with 0.2% sodium azide at 4°C until cryosectioning.

### Cryosectioning

Fixed spinal cords were mounted in optimal cutting temperature (OCT) compound (Agar Scientific, AGR1180) and sliced cross-sectionally in 30 μm-thick sections using a cryostat at −17°C. Each section was kept in antifreeze buffer (30% ethyleneglycol, 30% glycerol, 192.5 mM NaOH, 244 mM NaH$_2$PO$_4$) at 4°C until immunostaining.

### Immunohistochemistry of spinal cord sections

Two sections per lumbar L3-L6 segment were immunostained for ChAT, Calbindin, Parvalbumin, and GAD-67 at the same time per animal. This approach ensured systematic sampling across the lumbar enlargement, with an equal number of sections analysed from each anatomical level in every animal. Sampling based on anatomical levels was used instead of fixed section intervals because the mouse genotypes exhibit significant differences in body weight at this age, which could plausibly result in differences in spinal cord length. All steps were performed in Coplin jars with ~40 ml of solution, except for the antibody incubations and the antigen retrieval step. Briefly, the tissue was mounted onto Superfrost Plus glass microscope slides (Fisher, 10149870) then washed three times (15 min each) in

phosphate-buffered saline (PBS). Antigen retrieval was performed with sodium citrate buffer (10 mM Sodium citrate, 0.05% Tween 20, pH 6.0) prewarmed to 80–95°C for 30 minutes on a rocker at 30 osc/min. Permeabilization was performed with 0.05% Tween-20 (in PBS) for 10 minutes at 100 rpm (three times). Blocking was performed with PBS containing 0.1% Tween-20, 0.1M glycine, and 5% horse serum (Fisher, 11520516) for 90 minutes at 100 rpm. The sections were then probed overnight in a humidity chamber on an orbital shaker at 100 rpm and at 4°C in PBS containing the primary antibody solution (0.05% Tween-20, 1% horse serum) for polyclonal goat anit-ChAT (1:50 dilution, Merck/Sigma, AB144P), polyclonal guinea pig anti-calbindin D28k (1:200 dilution, Synaptic Systems, 214 004), polyclonal rabbit anti-parvalbumin (1:500 dilution, Synaptic Systems, 195 002), and polyclonal chicken anti-GAD67 (1:50 dilution, Bio-Techne, NBP1−02161). Following three 10-min PBS washes at 100 rpm, secondary probing was performed for 3 hours in the dark in PBS containing 0.05% Tween-20 and the relevant secondary antibodies for ChAT (1:250 dilution, donkey anti-goat IgG H + L Alexa Fluor® 405, Abcam, ab175664), Calbindin (1:500 dilution, Rhodamine TRITC-AffiniPure donkey anti-guinea pig Ig G H + L, Jackson Immuno Research, 706-025-148-JIR), Parvalbumin (1:500 dilution, Alexa Fluor 647-AffiniPure donkey anti-rabbit IgG H + L, Jackson Immuno Research, 711-605-152), and GAD-67 (1:500 dilution, donkey anti-chicken IgY H + L Highly Cross Adsorbed Alexa Fluor® 488, Life Technologies, A78948). Finally, four washes in PBS (15 min each) were performed at 100 rpm, and the slides were sealed with Invitrogen™ ProLong™ Glass Antifade Mountant (Fisher, 15820993).

## Image acquisition

Imaging was performed blinded to genotype. Images were acquired using a Leica SP8 confocal microscope with LAS X version 3.4 software. A 20x objective was used; however, since the entire spinal cord cross-section could not be captured in a single field of view, a tile scan acquisition was performed. Each tile of the spinal cord cross-section was imaged as a z-stack in four separate fluorescence channels, with each channel corresponding to a specific immunohistochemical marker: ChAT, Calbindin, Parvalbumin, or GAD-67. The individual tiles were then merged into a single mosaic image using the mosaic merge function of LAS X with statistical overlap blending.

## Image processing

Image analysis was performed blinded to genotype using custom-written ImageJ macros and MATLAB functions which we made publicly available [30]. The complete dataset analysed is also accompanying this publication (S1 File). First, maximum intensity projections were generated from z-stacks of each colour channel image to produce 2D representations of spinal cord cross-sections, allowing us to distinguish the different cell populations present in the tissue. Next, a region of interest (ROI) delineating the gray matter was manually drawn and saved for subsequent masking. The gray matter ROI was used to mask out non-gray matter areas. Then, a background subtraction step was implemented by employing a rolling ball algorithm with a radius of 20 pixels, which estimates and subtracts the smoothly varying background intensity from the image, enhancing the contrast of cellular structures. Manual thresholding was then applied to each individual channel image using the 'MaxEntropy' method which maximises the entropy between the dark (background) and bright (foreground) regions, to ensure accurate segmentation of the cells. The thresholded images were converted to binaries, and noise reduction followed by morphological closing were performed. Particle analysis was then used with cell type-specific size limits to extract cell centroids and boundaries. Specifically, ChAT-positive cells are expected to have a 10–40 μm diameter (area of 78.5–1256.6 μm$^2$) [31,32], Parvalbumin-positive cells a 10–20 μm diameter (area of 78.5–314 μm$^2$) [33], GAD-67-positive cells a 8–20 μm diameter (area of 50–314 μm$^2$) [34], and Calbindin-positive cells a 10–25 μm diameter (area of 78.5–490.9 μm$^2$) [35]. Additionally, when investigating ChAT-positive subpopulations, we divided them into interneurons of 10–25 μm diameter and motor neurons of 25–40 μm diameter.

## Colocalisation analysis

To quantify colocalisation, we loaded each cell's ROI—defined on marker A (e.g., ChAT)—and measured how much of that ROI overlapped thresholded pixels in the binary mask for marker B (e.g., GAD-67). We then computed the fraction of the ROI's area that was positive for marker B and only retained cells for which this overlap was ≥ 20%. This 20% cutoff was chosen to exclude spurious edge-touches or small contacts while still capturing genuine co-expression events.

## Cell distribution analysis

The central canal was manually delineated, and its centroid was computed. Subsequently, we computed the straight-line distance from each cell's centre to the central canal's centre, and we determined the orientation angle by using the inverse tangent of the differences in their horizontal and vertical positions. Finally, to adjust for differences in tissue size between samples, we normalised the cell distances relative to the gray matter. We did this by creating a binary mask from the saved gray matter ROI, computing a distance map that assigns each pixel a value based on its distance from the tissue edge, and then dividing each cell's distance (sampled from this map) by the maximum distance value to obtain normalised distances.

## Angle transformation for laterality correction

To ensure that cell distribution analyses were not confounded by arbitrary left-right orientations of free-floating spinal cord sections, we applied a transformation to the angle data. Angles were originally computed in ImageJ relative to the central canal, using the mathematical convention where 0° corresponds to the right, 90° to the dorsal side, 180° to the left, and 270° to the ventral side. Because the true anatomical left and right sides cannot be determined for each section (given the inherent symmetry of the tissue and the possibility of different mounting orientations) we transformed the angle data to eliminate any bias. Specifically, for each cell, if the angle was between 90° and 270° (indicating a location on the left side), it was transformed using: $\theta' = 180° - \theta$. Angles on the right side (0°–90° and 270°–360°) remained unchanged. This transformation effectively mirrored left-side angles onto the right side while maintaining dorsoventral positioning (Fig 1). This transformation ensured that left-right differences were removed, preventing the detection of artificial laterality due to arbitrary assignment of left and right.

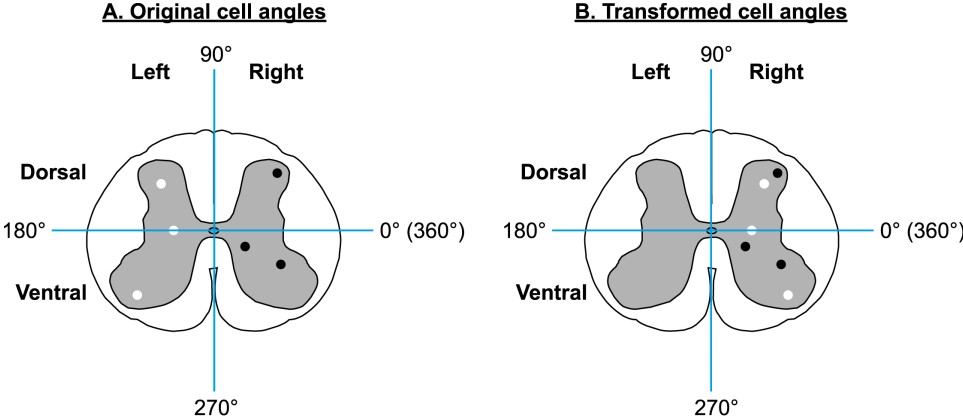

**Fig 1. Angle transformation for laterality correction.** Cells in the gray matter between 90° and 270° (left side; white cells) were mirrored onto the right side of the gray matter. The angle of cells originally on the right side (black cells) was not altered.

## Statistical analysis

Statistical analysis of linear data was performed using GraphPad Prism version 10. The Shapiro-Wilk test was used to confirm that all data were normally distributed and then an ordinary one-way analysis of variance (ANOVA) or a nested one-way ANOVA was used for the analysis of cell numbers and cell distances, respectively. No *post hoc* tests were used as none of the main effects were statistically significant at the 5% alpha level. Analysis of circular data was performed using the CircStat toolbox [36] in MATLAB R2024b. A circular median test (equivalent to a Kruskal-Wallis test for linear data) was used to compare cell angles between genotypes, followed by the Kuiper *post hoc* test, and adjusted for multiple comparisons using the False Discovery Rate (FDR) approach, specifically *via* the Benjamini–Hochberg procedure and through custom-written code which we made publicly available [30].

## Results

We immunostained spinal cord cross-sections from lumbar levels L3–L6 of mice with increasingly reduced dynein function and then performed confocal microscopy to identify different cell populations. An example of an immunostained spinal cord cross-section from each genotype is shown in Fig 2. First, we examined whether there were any changes in the numbers of different cell populations in the lumbar spinal cord of mice, by looking at each marker independently. We found no statistically significant differences in the number of ChAT⁺ cells between any of the genotypes (Fig 3A). This suggests there is no loss of cholinergic motor neurons in the lumbar spinal cord of mice with increasingly reduced dynein function. Previously, in a different set of 12 mice (three per genotype), we observed a non-significant trend towards cell loss with increasing dynein dysfunction, when counting Nissl-stained motor neurons (assumed but not confirmed to be ChAT⁺) specifically in the sciatic motor pool of the lumbar (L3–L6) spinal cord [25]. Here, the same non-significant trend was observed with the ChAT⁺ cells even without restricting the counting to the sciatic motor pool. This corroborates our prior conclusions that there is no significant motor neuron loss in the lumbar spinal cord of these mice, however, this could also be due to both studies being underpowered (i.e., only three biological replicates per genotype). It should be noted, however, that our analysis of ChAT⁺ cells was done globally on the entire gray matter, therefore, a small proportion (5–10%) of ChAT⁺ cells in Fig 3A are likely to be interneurons residing in the intermediate zone of the gray matter (lamina VII) and in regions bordering the central canal (lamina X). To investigate this, we divided the ChAT⁺ cells into two subpopulations based on their size, with interneurons represented by cells of 10–25 μm diameter (~78.5–491 μm² area) and motor neurons represented by cells of 25–40 μm diameter (491–1257 μm² area) [37]. As with the overall ChAT⁺ cells, this subpopulation analysis indicated no significant differences in the number of ChAT⁺ interneurons or motor neurons between genotypes, despite a trend towards a reduction in cell numbers with increasing dynein dysfunction (Fig 3B-C). Additionally, we also did not observe a significant loss of GAD-67⁺, Calbindin⁺, or Parvalbumin⁺ cells (Fig 3D-F), although, GAD-67⁺ cells showed the same non-significant trend towards a reduction of cell numbers with increasing dynein dysfunction as ChAT⁺ cells (Fig 3D).

The above single-marker analyses label broad classes, such as all cholinergic cells (ChAT⁺) or all GABAergic cells (GAD-67⁺), but many functionally distinct interneurons are defined by co-expression of two proteins. Therefore, we next investigated each double-positive population across genotypes to determine whether dynein impairment selectively compromises particular spinal cord microcircuits, even if overall single-marker counts remain unchanged. Firstly, a subset of cholinergic neurons (ChAT⁺) around the central canal (lamina X) co-express GAD-67, and these are thought to be cerebrospinal-fluid–contacting interneurons, which are sometimes called central autonomic or central cluster cells [38]. Secondly, almost all Parvalbumin⁺ neurons in the spinal cord are GABAergic and thus co-express GAD-67. These are typically called fast-spiking GABAergic interneurons. The two main populations include dorsal horn presynaptic inhibitory interneurons, which gate afferent input, and ventral Ia-inhibitory interneurons [39]. Finally, Parvalbumin and Calbindin are co-expressed in Renshaw cells, the classic recurrent-inhibitory interneurons in laminae VII/VIII, which is expected to make up only a small subset of all Parvalbumin⁺ cells [39]. In our tissues, we did not observe a significant difference in

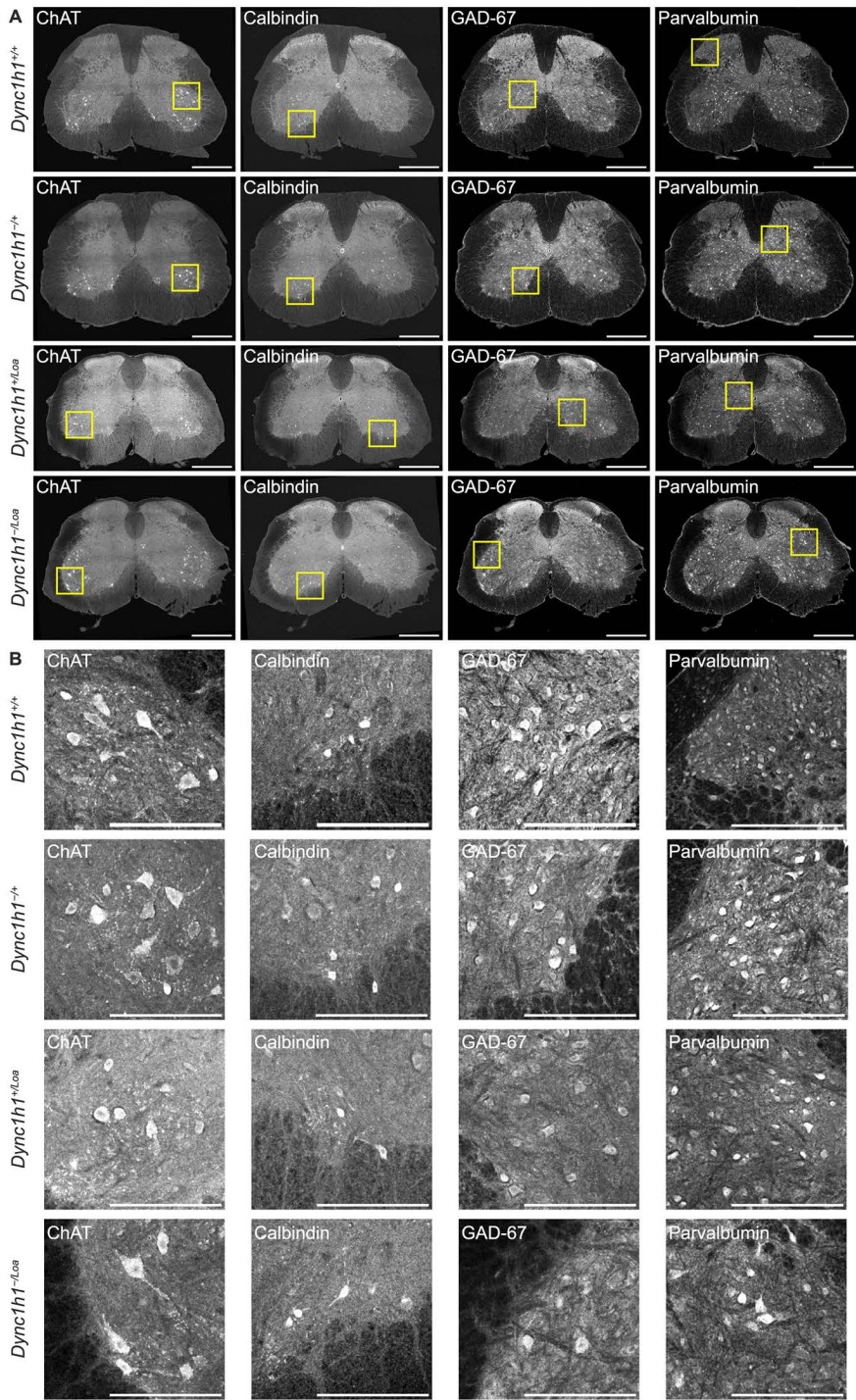

**Fig 2. Representative confocal microscopy images of immunostained spinal cord cross-sections from each genotype. (A)** Full mosaic-merge images. Images were adjusted for brightness and contrast for illustrative purposes only. **(B)** Corresponding insets from A to show examples of positively stained cells more clearly. Scale bars = 500 μm **(A)** or 200 μm **(B)**.

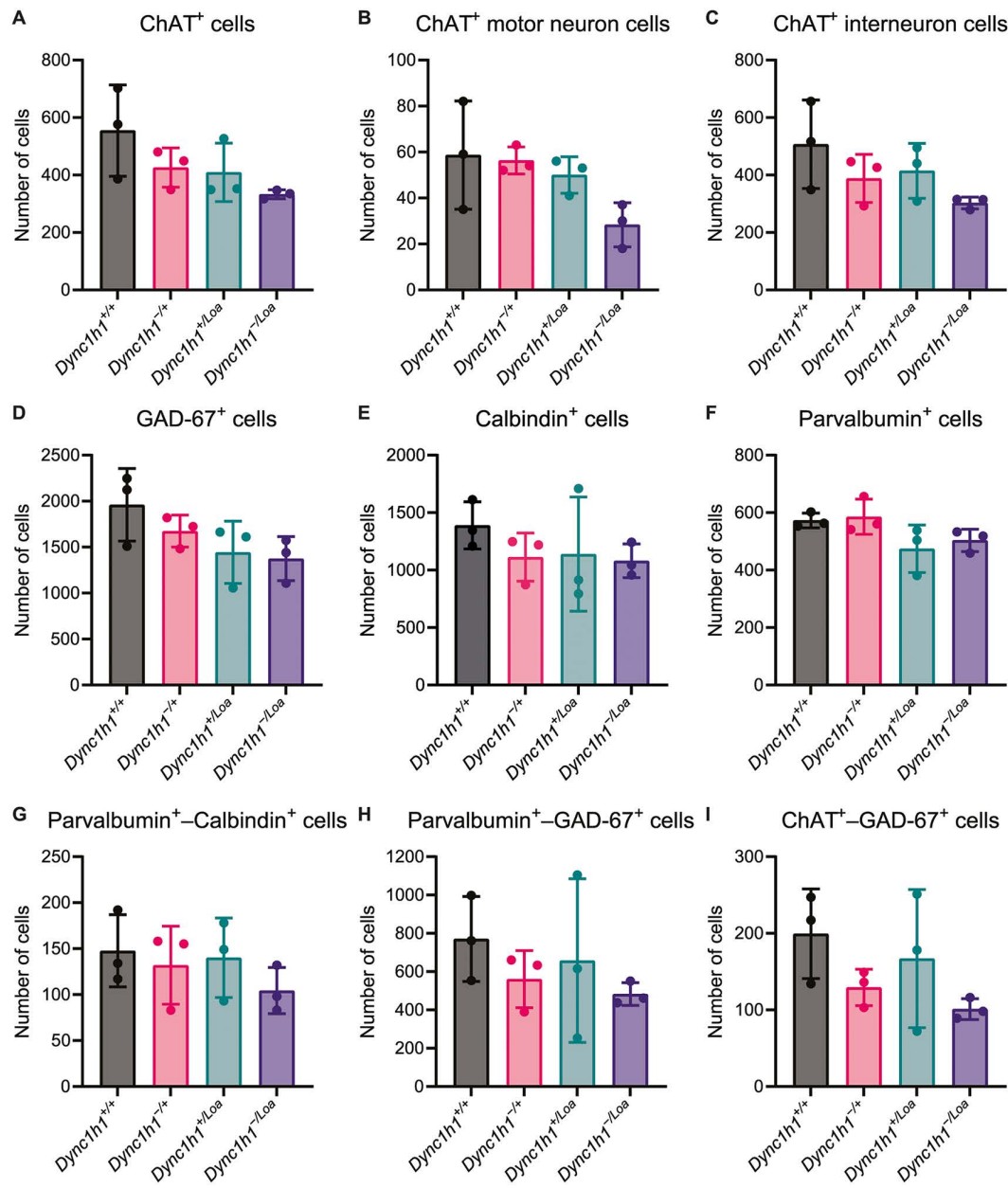

**Fig 3. Number of ChAT+ (A-C), GAD-67+ (D), Calbindin+ (E), Parvalbumin+ (F), Parvalbumin+−Calbindin+ (G), Parvalbumin+−GAD-67+ (H), and ChAT+−GAD-67+ (I) cells per biological replicate.** The number of cells analysed is the sum of immunostained positive cells from eight spinal cord cross-sections (two sections per lumbar level L3–L6) per biological replicate. Data are shown as mean±standard deviation and were analysed by an ordinary one-way ANOVA. N=3 biological replicates per genotype.

the number of any of these double-positive cell populations between genotypes (Fig 3G-I). This suggests that impairing dynein function does not produce overt loss of any of the major spinal interneuron populations or their more restricted double-positive subsets.

Together, these findings indicate that reductions in dynein activity, even when sufficient to drive clear motor deficits in these mice, are not accompanied by statistically significant cell death of either motor neurons or key inhibitory interneuron

classes in the lumbar spinal cord. Functionally, this preservation of cell number suggests that the motor dysfunction seen in dynein-deficient animals is more likely to arise from impairments in axonal transport, synaptic connectivity, or neuronal physiology, rather than from degeneration of spinal neurons themselves. Future studies employing electrophysiological recordings and synaptic ultrastructure analyses will be needed to determine whether hypomorphic dynein perturbs circuit activity or synaptic maintenance in the absence of cell death.

Next, we assessed the distribution of different cell types within the gray matter. The central canal is a well-defined, anatomically conserved structure in the spinal cord, and therefore, we used it as a reference point to standardise measurements across samples. We first calculated the distance of each immunopositive cell from the central canal and normalised it to the gray matter boundaries to ensure that any observed differences were not simply due to variability in overall tissue size. The distance from the central canal can reveal gradients in cell positioning. Certain interneuron populations might cluster close to the central canal, while others are preferentially located in more peripheral regions of the gray matter. Such gradients can suggest differences in developmental origins or migratory cues [40]. In our tissues, we did not observe any statistically significant differences in the distances of cells from the central canal between genotypes, for any of the cell markers we used, either independently (Fig 4A-F) or double-positive (Fig 4G-I). This suggests that any dynein dysfunction does not appear to disrupt the spatial organisation of spinal neurons in the lumbar enlargement. In other words, both single-marker and co-labelled cell populations occupy their normal positions relative to the central canal regardless of genotype. This preservation of positional gradients implies that impaired dynein does not affect the ability of the cells to follow the migratory cues and microenvironmental signals guiding interneuron and motor neuron placement during development.

In addition to measuring the radial distance of cells from the central canal, we analysed the angular distribution of each cell relative to this central landmark. The angle provides directional information, delineating which areas of the spinal cord the cells occupy. This analysis allowed us to determine whether cells exhibit a preferential orientation, such as a bias toward dorsal or ventral regions, independent of their distance from the central canal. For example, this is already known to be the case for cholinergic motor neurons, which should predominantly be in the ventral horns [41]. Due to the inherent circularity of the angle data, where 0° is equivalent to 360°, standard linear plots like bar graphs and violin plots are not suitable for accurately representing angle measurements. Thus, we employed circular-specific visualisations (i.e., rose diagrams) to display the directional distribution of cell orientations across all lumbar levels together. As expected, we observed most ChAT+ cells (likely motor neurons) to be in the ventral horns (between 270° and 360°), whereas a small proportion (likely interneurons) was located around the central canal (Fig 5A). The angular distribution of Calbindin+, GAD-67+, and Parvalbumin+ cells were also visualised (Fig 5B-D). Similar visualisations were created for double-positive cells (Fig 6A-C).

Statistical analysis using a circular median test (equivalent to a Kruskal-Wallis test for linear data) to compare cell angles between genotypes (ignoring lumbar level) revealed a significant effect of genotype for ChAT+ cells (p < 0.0001). Further subdivision of the ChAT+ cells into interneuron and motor neuron subpopulations based on cell size as before, revealed that the significance remained in the interneuron but not the motor neuron subpopulation. *Post hoc* tests for ChAT+ cells (total) and ChAT+ interneurons were done using the circular-data specific Kuiper test and adjusted for multiple comparisons using the False Discovery Rate (FDR) approach, specifically *via* the Benjamini–Hochberg procedure. This revealed that the significant differences in the angular distribution of ChAT+ cells/interneurons were between all possible pairwise genotype comparisons (all p < 0.0001) except for *Dync1h1*+/Loa versus *Dync1h1*−/Loa (not significant; Fig 7A-B). The angles of GAD-67+ cells similarly displayed a significant effect of genotype (p < 0.0001), and the *post hoc* pairwise comparisons revealed this to be between all possible pairwise genotype comparisons (p = 0.033 for *Dync1h1*+/Loa versus *Dync1h1*−/Loa and p < 0.0001 for all others; Fig 7C). Furthermore, the circular median test for ChAT+−GAD-67+ and Parvalbumin+−GAD-67+ double-positive cells also displayed a statistically significant effect of genotype (p < 0.0001 and p < 0.05, respectively). *Post hoc* tests revealed that the significant differences were again between all possible pairwise genotype

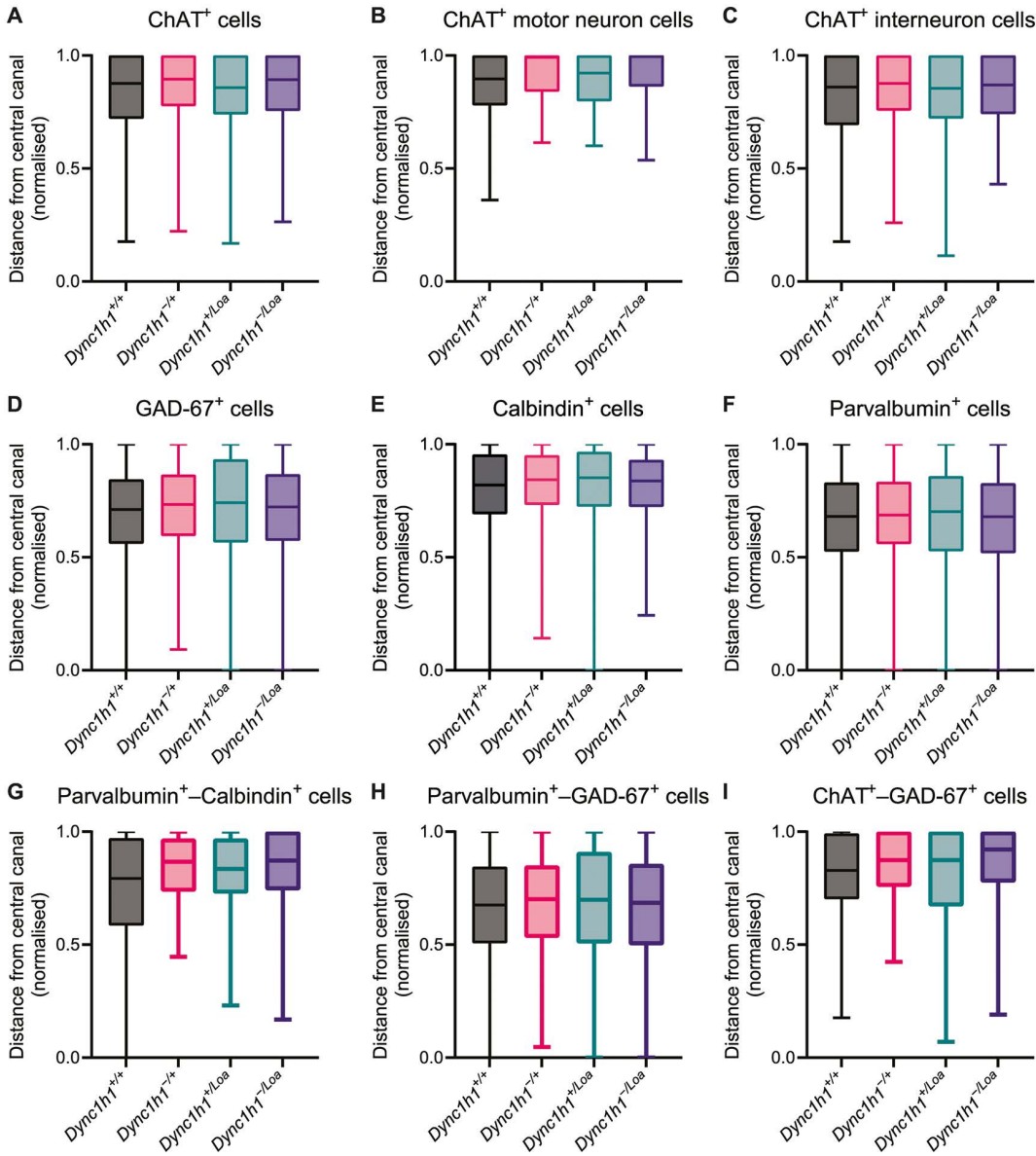

**Fig 4. (A-I) Normalised distances of cells relative to the central canal.** The distances were normalised to the size of the gray matter to account for size differences between spinal cord cross-sections. The normalised distance is a unitless fraction between 0 (closest to the central canal) and 1 (farthest from the central canal). Data are from eight spinal cord cross-sections (two sections per lumbar level L3–L6) per biological replicate. Boxes represent the interquartile range with a line in the middle representing the median. The whiskers represent the minimum and maximum values. Data were analysed by a nested one-way ANOVA. N = 3 biological replicates per genotype.

comparisons (Fig 7D-E). The angles of Calbindin+, Parvalbumin+, and Parvalbumin+−Calbindin+ double-positive cells did not display a statistically significant effect of genotype (p = 0.2963, p = 0.1588, and p = 0.1453, respectively) and thus no *post hoc* tests were conducted for these.

It is important to interpret these statistically significant differences in context. Although the circular statistics revealed genotype differences for ChAT+, GAD-67+, ChAT+−GAD-67+, and Parvalbumin+−GAD-67+ cells, the actual shifts in mean

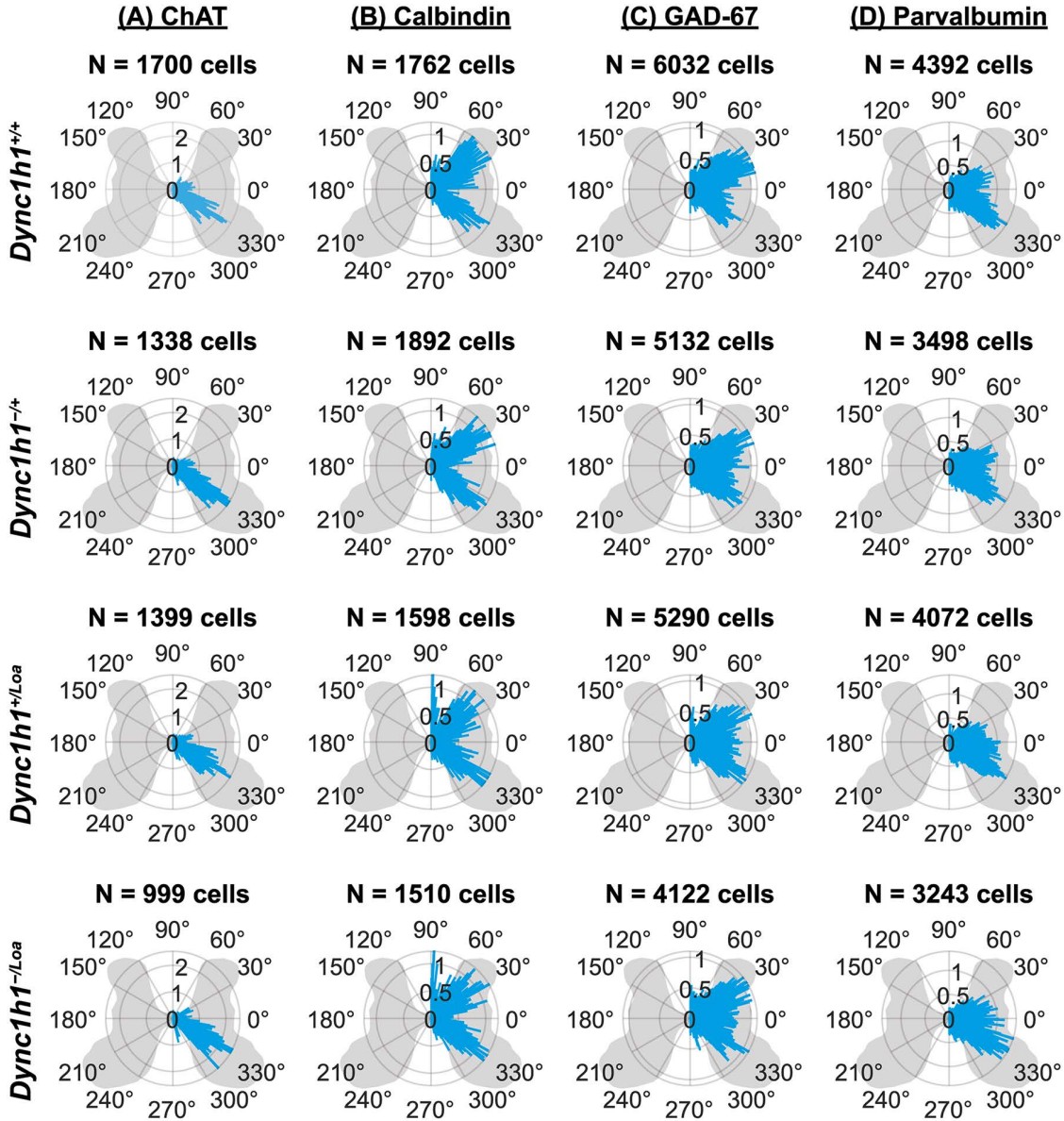

**Fig 5. (A-D) Spatial distribution of immunopositive cells displayed as rose diagrams (i.e., circular histograms) of cell angles relative to the central canal.** All left side angles were mirrored to the right side to avoid displaying artificial laterality due to the inability to distinguish the true left and right sides of free-floating spinal cord cross-sections. The length of each bar represents the percentage of cells found at a particular angle (not the distance from the central canal; distance information is ignored in these plots). Data are from eight spinal cord cross-sections (two sections per lumbar level L3–L6) per biological replicate. N = 3 biological replicates per genotype.

angle remain modest (only 5–10°), and the variability within groups is low, as indicated by the small circular variance values in Table 1. Such tiny angular displacements translate to minimal lateral shifts, on the order of a few tens of microns along a roughly 300 µm–radius gray-matter arc, which are well within the width of a single laminar band. Moreover, both ChAT+–GAD-67+ cerebrospinal-fluid–contacting interneurons and Parvalbumin+–GAD-67+ fast-spiking interneurons occupy highly discrete niches, and thus a shift of only a few degrees (10–20 µm) would not meaningfully alter their synaptic

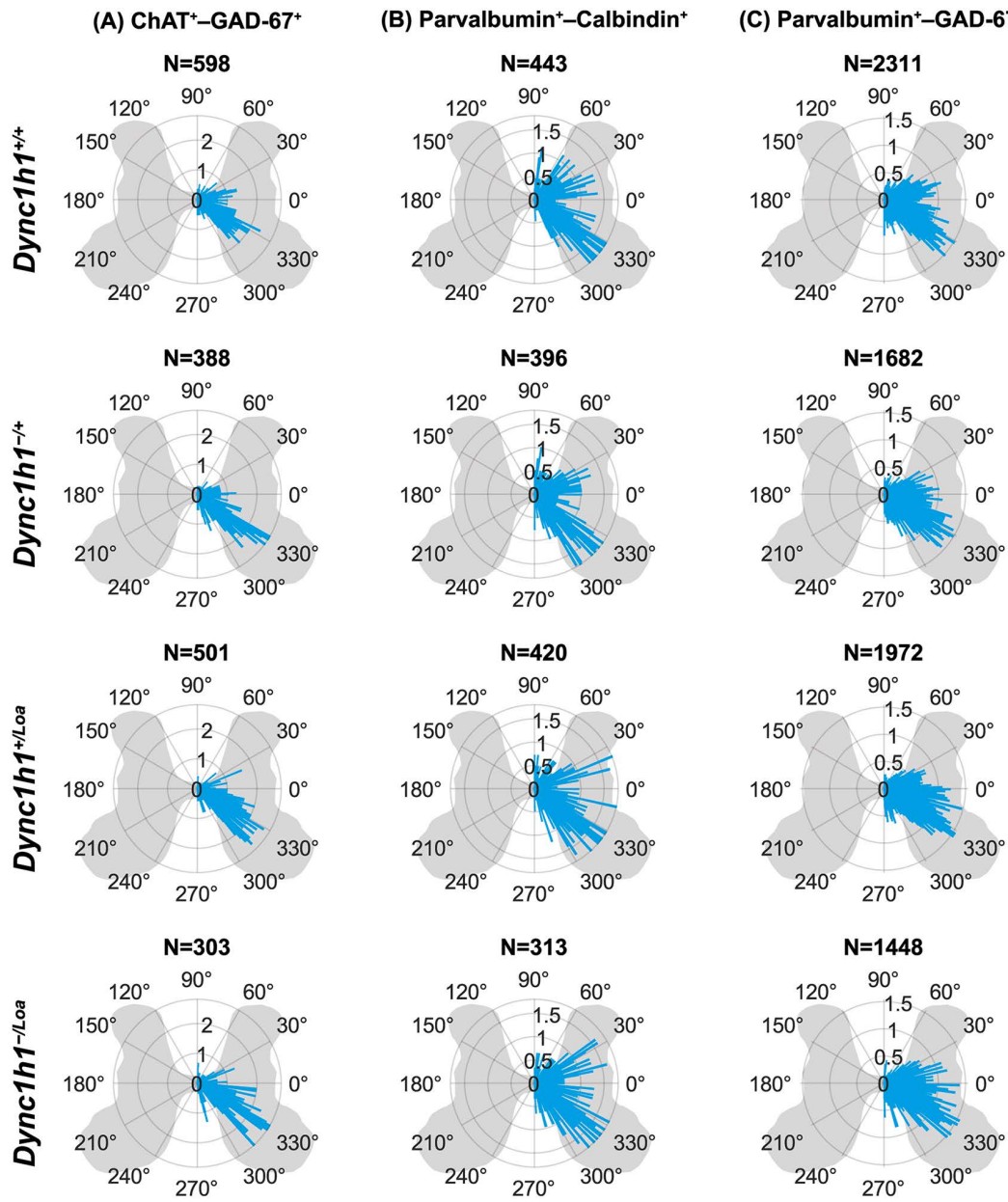

**Fig 6. (A-C) Spatial distribution of double-positive cells displayed as rose diagrams (i.e., circular histograms) of cell angles relative to the central canal.** All left side angles were mirrored to the right side to avoid displaying artificial laterality due to the inability to distinguish the true left and right sides of free-floating spinal cord cross-sections. The length of each bar represents the percentage of cells found at a particular angle (not the distance from the central canal; distance information is ignored in these plots). Data are from eight spinal cord cross-sections (two sections per lumbar level L3–L6) per biological replicate. N = 3 biological replicates per genotype.

partners or circuit roles. Therefore, despite the mathematically significant differences, their magnitude is not large enough to be of clear biological consequence. Overall, while the statistical analysis identifies significant differences, careful consideration of effect size and the biological context is required to determine if these differences have meaningful implications for spinal cord organisation.

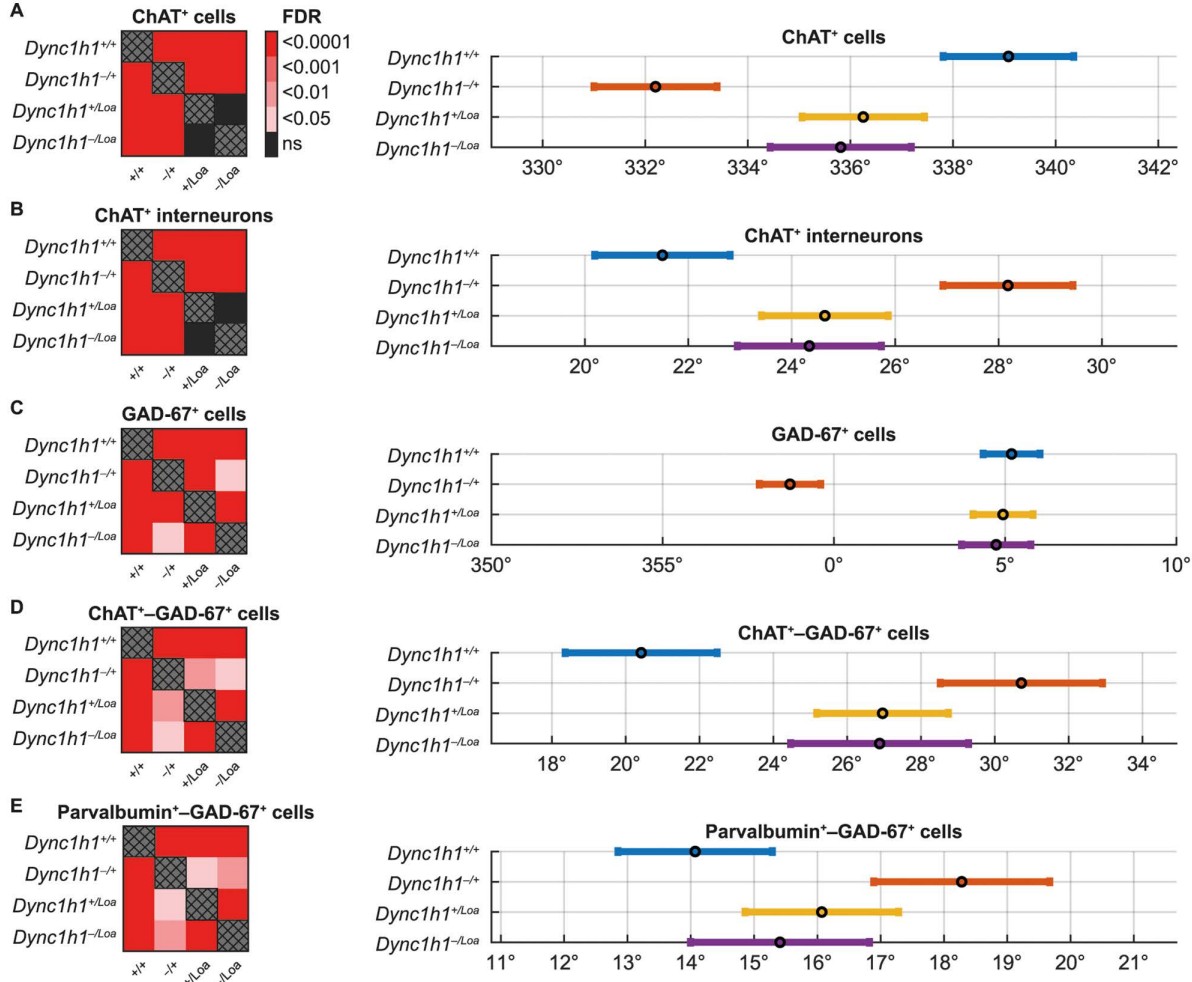

**Fig 7. (A-E) Summary of *post hoc* pairwise comparisons for cell types that displayed significant main effects.** The left column shows heatmaps of false discovery rate (FDR) values to indicate statistical significance. The right column shows the mean angle of the cells (circles) and the 95% confidence interval (error bars) for each genotype.

## Discussion

The goal of this study was to determine whether impaired dynein function alters the survival or positioning of spinal interneurons in a mouse model of ALS-like pathology. Contrary to our original hypothesis, we observed no significant loss of any major interneuron population, including cholinergic, GABAergic, parvalbumin- and calbindin-positive cells, across genotypes with progressively reduced dynein activity. Subdividing ChAT⁺ cells into presumed motor neurons versus smaller-diameter interneurons likewise revealed no genotype-dependent differences in cell number. Furthermore, the radial distribution of every marker, quantified as the normalised distance from the central canal, was preserved, indicating that gross migratory defects or laminar misplacement do not accompany reduced dynein function in these mice.

Functionally, maintaining the correct laminar and radial distributions of the cells we investigated is critical for proper circuit assembly and synaptic connectivity. Therefore, our data suggest that the motor impairments observed in dynein-deficient mice are unlikely to stem from gross mislocalisation of neurons within the gray matter. Instead, they reinforce the view that the primary contributions of dynein to motor performance hinge on its roles in intracellular transport and

**Table 1. Circular means and variances from cell angle data for cell types displaying statistically significant differences between genotypes.**

| Genotype | Cell type | Circular mean (degrees) | Circular variance (degrees) |
|---|---|---|---|
| Dync1h1 +/+ | ChAT+ | 339.08 | 0.1790 |
| | GAD-67+ | 5.19* | 0.2472 |
| | ChAT+ interneurons | 21.50 | 0.1710 |
| | ChAT+−GAD-67+ | 20.42 | 0.1675 |
| | Parvalbumin+−GAD-67+ | 14.07 | 0.2141 |
| Dync1h1 −/+ | ChAT+ | 332.2 | 0.1323 |
| | GAD-67+ | 358.72** | 0.2462 |
| | ChAT+ interneurons | 28.18 | 0.1271 |
| | ChAT+−GAD-67+ | 30.72 | 0.1294 |
| | Parvalbumin+−GAD-67+ | 18.28 | 0.2049 |
| Dync1h1 +/Loa | ChAT+ | 336.25 | 0.1361 |
| | GAD-67+ | 4.94* | 0.2433 |
| | ChAT+ interneurons | 24.64 | 0.1290 |
| | ChAT+−GAD-67+ | 26.96 | 0.1124 |
| | Parvalbumin+−GAD-67+ | 16.07 | 0.1872 |
| Dync1h1 −/Loa | ChAT+ | 335.81 | 0.1303 |
| | GAD-67+ | 4.74* | 0.2508 |
| | ChAT+ interneurons | 24.34 | 0.1226 |
| | ChAT+−GAD-67+ | 26.88 | 0.1273 |
| | Parvalbumin+−GAD-67+ | 15.41 | 0.1867 |

Data for ChAT+ motor neurons only, Calbindin+, Parvalbumin+, and Parvalbumin+−Calbindin+ double-positive cells are omitted as they did not display significant main effects. Note that this is circular data and therefore an angle of 0° is indistinguishable from 360°. This is important when comparing the value marked with ** to the values marked with *, as they are essentially all very close in the circular scale.

synaptic maintenance rather than on the initial positioning or survival of spinal cord neurons. However, several studies have demonstrated that compromising the motor activity of dynein (*Loa* mutation), expression (RNAi), cargo–linkage *via* its accessory proteins (BICD2), or its coding sequence in humans (multiple *DYNC1H1* variants) consistently produces radial migration delay, abnormal cortical lamination, or overt malformations of cortical development [18,22,42–46]. The mechanistic consensus is that dynein-driven nuclear movement, both inter-kinetic in progenitors and somal translocation in neurons, is indispensable for building a correctly layered cerebral cortex, therefore, there is a mismatch between the brain and spinal cord effects of impaired dynein function, based on our current findings.

We also tested whether dynein impairment subtly shifts the angular distribution of cells. While circular-statistical analyses uncovered statistically significant genotype effects for ChAT+, GAD-67+, and certain double-positive populations, the measured shifts were extremely modest (5–10°) along an arc of ~300 μm, which are distances well within a single laminar band, and are thus unlikely to alter circuit connectivity in any meaningful way.

Functionally, the preservation of both cell numbers and macro-anatomical positioning argues against interneuron death or gross mislocalisation as drivers of the motor deficits in dynein-deficient animals. Instead, our data reinforce a model in which the primary contributions of dynein to neuromuscular dysfunction likely arise from disrupted intracellular transport, synaptic maintenance, and protein homeostasis, phenomena we have previously linked to TDP-43 aggregation and p62 upregulation in these same mouse lines [25]. Indeed, interneuron hyperexcitability and synaptic disconnection, rather than cell loss per se, have been implicated in ALS pathogenesis by prior studies of both motor and non-motor circuits [5,8,27–29].

There are important limitations to note. First, our analysis sampled three biological replicates per genotype with eight sections each; the resulting cell counts, while substantial, may still lack power to detect very small population changes. Therefore, the present findings should be interpreted as indicating that no gross interneuron loss or major spatial disorganisation is present in these models at 52 weeks of age. Detecting smaller effects would require larger cohorts and additional analyses, which were beyond the scope of the present study. Second, we focused exclusively on classical inhibitory markers (GAD-67, parvalbumin, calbindin) and cholinergic cells; other interneuron subtypes (e.g., glycinergic, somatostatin-expressing) or excitatory interneurons were not assessed. Third, all measurements were made at 52 weeks of age, potentially missing dynamic developmental or early-disease alterations in migration or survival. Finally, our image-processing pipeline relied on thresholding and size-based segmentation, which may misclassify cells near the chosen parameter boundaries.

Future work should therefore extend these structural analyses to functional assays such as patch-clamp recordings or calcium imaging of identified interneurons, *in vivo* transport assays to directly measure dynein-dependent cargo movement, and connectomic studies to assess synapse integrity. Examining earlier developmental time points, incorporating lineage-tracing of migrating precursors, and applying single-cell transcriptomics could reveal transient or subtype-specific vulnerabilities. Finally, conditional ablation of dynein subunits in interneuron populations would help dissect cell-autonomous from non-cell-autonomous contributions to ALS-like phenotypes.

In summary, impaired dynein function in cholinergic motor pools does not precipitate cell death or gross mislocalisation of key spinal interneurons. Rather, our findings shift the focus toward the essential roles of dynein in maintaining synaptic connectivity and proteostasis, underscoring the importance of functional rather than structural disruptions in ALS pathogenesis.

## Supporting information

**S1 File. The complete dataset that was statistically analysed.** It includes the transformed angles and normalised distances of each immunopositive cell, relative to the central canal.
(XLSX)

## Author contributions

**Conceptualization:** Peter Nigel Leigh, Majid Hafezparast.

**Data curation:** Eleni Christoforidou.

**Formal analysis:** Eleni Christoforidou, Jordan S. Rowe.

**Funding acquisition:** Eleni Christoforidou, Peter Nigel Leigh, Majid Hafezparast.

**Investigation:** Eleni Christoforidou, Jordan S. Rowe, Fabio A. Simoes.

**Methodology:** Eleni Christoforidou.

**Project administration:** Eleni Christoforidou.

**Resources:** Raphaelle Cassel, Luc Dupuis.

**Software:** Eleni Christoforidou.

**Supervision:** Majid Hafezparast.

**Visualization:** Eleni Christoforidou.

**Writing – original draft:** Eleni Christoforidou.

**Writing – review & editing:** Eleni Christoforidou, Jordan S. Rowe, Fabio A. Simoes, Raphaelle Cassel, Luc Dupuis, Peter Nigel Leigh, Majid Hafezparast.

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
