## [Decision Letter · Decision Letter 0]

29 Jan 2026

Dear Dr. Hafezparast,

Thank you for submitting your manuscript to PLOS ONE. After careful consideration, we feel that it has merit but does not fully meet PLOS ONE’s publication criteria as it currently stands. Therefore, we invite you to submit a revised version of the manuscript that addresses the points raised during the review process.

We look forward to receiving your revised manuscript.

Kind regards,

Xiaona Wang, Ph.D

Academic Editor

PLOS One

Journal Requirements:

2. To comply with PLOS One submissions requirements, in your Methods section, please provide additional information regarding the experiments involving animals and ensure you have included details on (1) methods of sacrifice, (2) methods of anesthesia and/or analgesia, and (3) efforts to alleviate suffering.

“Motor Neurone Disease Association”

Reviewers' comments:

Reviewer's Responses to Questions

**Comments to the Author**

1. Is the manuscript technically sound, and do the data support the conclusions?

Reviewer #1: Yes

Reviewer #2: Partly

2. Has the statistical analysis been performed appropriately and rigorously?

Reviewer #1: Yes

Reviewer #2: No

3. Have the authors made all data underlying the findings in their manuscript fully available?

Reviewer #1: Yes

Reviewer #2: Yes

4. Is the manuscript presented in an intelligible fashion and written in standard English?

Reviewer #1: Yes

Reviewer #2: Yes

Reviewer #1: Overall this is a nice short follow up paper to the main mouse model paper detailing their mutant Dynein lines. This one focuses on interneurons and whether there are any differences between the 4 mouse lines in terms of number or placement of these. There is no real difference and the only real change is an angle change for interneurons though it is hard to understand the significance of this. The major issue with this paper is that it is underpowered but given the lack of any major differences, i am not sure adding more animals in is ethical. There is not much to this paper and it is based on one set of images with different analyses but it overall adds to the field anyway.

Minor comments:

Please include a section detailing the mouse phenotypes. Although you give some details in the penultimate paragraph of the introduction, these are very generic and it is not clear at what age the symptoms start or the difference becomes significant from NTg littermates and so it is unclear why 52 weeks was selected as the time point for the tissue collection. Please add this all in so that someone can read this paper without having to read the other one as well

Regarding the animals, there is no information on the sex of the animals that were used for this study and this feels important when considering things like angles in spinal cord. If only one sex was used, please make it clear why and justify it.

Page 14, lines 331-334. please put in figure references for this section.

Reviewer #2: The manuscript is clearly motivated by the hypothesis that hypomorphic dynein function in cholinergic (ChAT+) neurons disrupts the development, survival, or spatial positioning of inhibitory interneuron populations in the lumbar spinal cord. The work ultimately presents a largely negative anatomical outcome: despite robust ALS-like phenotypes previously reported in these dynein-impaired models, the lumbar spinal cord at 52 weeks shows no evidence of inhibitory interneuron loss or overt, large-scale defects in interneuron positioning/migration.

Mechanistically, it is important to note that the genetic manipulation selectively reduces dynein function in ChAT+ neurons (motor neurons and cholinergic interneurons), whereas the primary outcome measures quantify broader inhibitory and marker-defined populations (GAD-67+, parvalbumin+, calbindin+, and their overlaps). Because these inhibitory subsets are not necessarily within the ChAT lineage, the experiments primarily test indirect (non–cell-autonomous) consequences of cholinergic dynein dysfunction on inhibitory interneuron survival and spatial organization, rather than strictly cell-autonomous effects within the manipulated population. This distinction should be made explicit in the framing and interpretation.

A major limitation is the absence of a power analysis together with the very small biological sample size (N = 3 mice/genotype). With this N, the study is likely powered only to detect large effects, and “no significant difference” cannot be taken as evidence that no biologically meaningful effect exists—particularly for smaller interneuron subpopulations and double-positive subsets, where counts may be low and variance comparatively high. Given that the central conclusion is essentially null, increasing biological replication would substantially strengthen the manuscript. While the current dataset can reasonably support the conservative statement that no gross interneuron loss or large-scale mispositioning was observed, it is underpowered to exclude more modest but potentially meaningful differences. If feasible, expanding to approximately 5–6 mice per genotype would improve confidence intervals and stabilize variance estimates, and would likely reduce reviewer concern about interpreting negative results. To support stronger claims that there is no biologically meaningful interneuron loss or mispositioning, a target of ≥6 mice/genotype (or more if variability is high, or if double-positive subsets are rare) is recommended.

Methodological reporting should also be strengthened. Cell counts derived from confocal images are appropriate for the question, but the Methods need to provide sufficient detail to convince readers that sampling and quantification were unbiased and comparable across genotypes. In particular, it should be stated explicitly whether counts were performed bilaterally, how counting boundaries/regions of interest were defined, and whether the sampling scheme was systematic and evenly spaced across L3–L6. Finally, key biological and classification details should be included—most notably the sex of the mice and whether groups were balanced, given known sex effects in ALS-like phenotypes and spinal cord measures.

Overall, I recommend major revision. The question is worthwhile and the dataset may be publishable, but the current sample size/power and statistical framing do not yet support the strength of the null conclusions, and additional methodological clarity (and ideally increased N) would substantially improve the rigor and interpretability of the work.

.

Reviewer #1: No

Reviewer #2: **Yes:**Gulgun SengulGulgun SengulGulgun SengulGulgun Sengul

---

## [Author Response · Author response to Decision Letter 1]

13 Mar 2026

Dear Editor,

We would like to thank you and the reviewers for the careful evaluation of our manuscript and for the constructive comments and suggestions. We have revised the manuscript accordingly and believe that these changes have improved the clarity, methodological transparency, and interpretation of the study. In particular, we have expanded the description of the mouse phenotypes and the rationale for the selected timepoint, clarified the genetic framework and interpretation of the findings, strengthened the discussion of limitations related to sample size and statistical power, and improved the reporting of methodological details regarding sampling and quantification procedures. All changes made to the manuscript are indicated in the tracked-changes version, and line numbers referenced in the responses below correspond to the tracked-changes file.

Response to the Editor

Author’s response and action taken:

The revised manuscript now meets the style requirements.

2. To comply with PLOS One submissions requirements, in your Methods section, please provide additional information regarding the experiments involving animals and ensure you have included details on (1) methods of sacrifice, (2) methods of anesthesia and/or analgesia, and (3) efforts to alleviate suffering.

Author’s response and action taken:

Methods of sacrifice, anaesthesia, and analgesia are stated in lines 181-183. Efforts to alleviate suffering are stated in lines 169-175.

“Motor Neurone Disease Association”

Author’s response and action taken:

The funders had no role in study design, data collection and analysis, decision to publish, or preparation of the manuscript. This has been confirmed in the cover later.

Author’s response and action taken:

This work was funded by the Motor Neurone Disease Association grants Christoforidou/Nov24/2494-793 (to EC, MH, and PNL) and Dupuis/Apr16/852-791(to LD and MH). The portal system doesn’t have an option for inputting this information. Following the advice from the Peer Review Operations Specialist, we have provided this Financial Disclosure statement in the author comments field.

Author’s response and action taken:

We have clarified the ethics statement in the Methods section (lines 155-160). This study involved no human participants, and therefore informed consent was not applicable, and is not stated in the manuscript.

Author’s response and action taken:

We have now included this caption in the revised manuscript, in lines 688-691, and cited it in the text in lines 241-242.

Author’s response and action taken:

The reviewers did not recommend citing any additional works.

Response to Reviewers

Reviewer #1: Overall this is a nice short follow up paper to the main mouse model paper detailing their mutant Dynein lines. This one focuses on interneurons and whether there are any differences between the 4 mouse lines in terms of number or placement of these. There is no real difference and the only real change is an angle change for interneurons though it is hard to understand the significance of this. The major issue with this paper is that it is underpowered but given the lack of any major differences, I am not sure adding more animals in is ethical. There is not much to this paper and it is based on one set of images with different analyses but it overall adds to the field anyway.

Minor comments:

Please include a section detailing the mouse phenotypes. Although you give some details in the penultimate paragraph of the introduction, these are very generic and it is not clear at what age the symptoms start or the difference becomes significant from NTg littermates and so it is unclear why 52 weeks was selected as the time point for the tissue collection. Please add this all in so that someone can read this paper without having to read the other one as well.

Author’s response and action taken:

To address this, we have expanded the Introduction section to describe the onset and progression of the phenotypes observed in these mouse strains, including the timing at which differences from wildtypes become apparent. This is in lines 111-123. We also clarify that 52 weeks was selected as the tissue collection time point because it represents a late adult stage at which the motor phenotype and associated pathological changes are well established, allowing us to examine whether long-term dynein dysfunction leads to loss or mislocalisation of spinal interneurons. This is in lines 136-139.

Regarding the animals, there is no information on the sex of the animals that were used for this study and this feels important when considering things like angles in spinal cord. If only one sex was used, please make it clear why and justify it.

Author’s response and action taken:

We have now clarified that only male mice were used in this study. This is sated in lines 28 and 178-181. Male mice were selected for analysis to maintain consistency with our previous characterisation of this mouse model, in which sex-dependent differences were observed in several phenotypic measures, including body weight, grip strength decline, and neuromuscular junction morphology. In particular, the progression of motor deficits was more pronounced in male mice, whereas female mice exhibited a milder or partially divergent phenotype. To reduce biological variability and ensure comparability with these previously reported phenotypes, the current study therefore focused on male animals.

Page 14, lines 331-334. please put in figure references for this section.

Author’s response and action taken:

We have now added this in line 372.

Reviewer #2: The manuscript is clearly motivated by the hypothesis that hypomorphic dynein function in cholinergic (ChAT+) neurons disrupts the development, survival, or spatial positioning of inhibitory interneuron populations in the lumbar spinal cord. The work ultimately presents a largely negative anatomical outcome: despite robust ALS-like phenotypes previously reported in these dynein-impaired models, the lumbar spinal cord at 52 weeks shows no evidence of inhibitory interneuron loss or overt, large-scale defects in interneuron positioning/migration.

Mechanistically, it is important to note that the genetic manipulation selectively reduces dynein function in ChAT+ neurons (motor neurons and cholinergic interneurons), whereas the primary outcome measures quantify broader inhibitory and marker-defined populations (GAD-67+, parvalbumin+, calbindin+, and their overlaps). Because these inhibitory subsets are not necessarily within the ChAT lineage, the experiments primarily test indirect (non–cell-autonomous) consequences of cholinergic dynein dysfunction on inhibitory interneuron survival and spatial organization, rather than strictly cell-autonomous effects within the manipulated population. This distinction should be made explicit in the framing and interpretation.

Author’s response and action taken:

We now explicitly state that the study examines potential non–cell-autonomous effects of dynein dysfunction in cholinergic neurons on inhibitory interneuron populations in the lumbar spinal cord. This is in lines 139-147.

A major limitation is the absence of a power analysis together with the very small biological sample size (N = 3 mice/genotype). With this N, the study is likely powered only to detect large effects, and “no significant difference” cannot be taken as evidence that no biologically meaningful effect exists—particularly for smaller interneuron subpopulations and double-positive subsets, where counts may be low and variance comparatively high. Given that the central conclusion is essentially null, increasing biological replication would substantially strengthen the manuscript. While the current dataset can reasonably support the conservative statement that no gross interneuron loss or large-scale mispositioning was observed, it is underpowered to exclude more modest but potentially meaningful differences. If feasible, expanding to approximately 5–6 mice per genotype would improve confidence intervals and stabilize variance estimates, and would likely reduce reviewer concern about interpreting negative results. To support stronger claims that there is no biologically meaningful interneuron loss or mispositioning, a target of ≥6 mice/genotype (or more if variability is high, or if double-positive subsets are rare) is recommended.

Author’s response and action taken:

We agree that studies aiming to detect small effect sizes benefit from larger numbers of biological replicates. However, this study was designed as an initial anatomical assessment to identify whether substantial changes in interneuron populations were present. The conclusions of the manuscript have been carefully framed in conservative terms. Specifically, we state that no gross interneuron loss or large-scale changes in spatial distribution were observed, rather than concluding that more subtle differences cannot occur. Increasing the number of animals to improve power for detecting small differences would require the use of additional animals. In accordance with the principle of Reduction within the 3Rs framework for animal research, and given that the current results did not indicate large or systematic changes in interneuron populations, we considered that further animal use was not ethically justified for the scope of the present study. Additionally, the number of animals used in this work was determined by the resources and funding available for the project at the time the experiments were performed. We have now clarified the limitations of the sample size in the Discussion and explicitly state that the present study cannot exclude smaller changes in interneuron subpopulations. Future studies with larger cohorts or complementary approaches will be required to assess more subtle effects. This is in lines 543-548.

Methodological reporting should also be strengthened. Cell counts derived from confocal images are appropriate for the question, but the Methods need to provide sufficient detail to convince readers that sampling and quantification were unbiased and comparable across genotypes. In particular, it should be stated explicitly whether counts were performed bilaterally, how counting boundaries/regions of interest were defined, and whether the sampling scheme was systematic and evenly spaced across L3–L6.

Author’s response and action taken:

The information regarding sampling and quantification procedures was already described in the Methods section, although we recognise that these details were distributed across several subsections and may not have been immediately apparent.

With regards to whether counts were performed bilaterally and how counting boundaries/regions of interest were defined:

Full spinal cord cross-sections were captured using confocal tile scans and merged into mosaic images prior to analysis, as we already stated in the Methods (lines 232-237). Additionally, in the “Image processing” section we state that “a region of interest (ROI) delineating the gray matter was manually drawn and saved for subsequent masking” (lines 245-246), ensuring that analyses were restricted to the entire gray matter regions of the spinal cord. All immunopositive cells within the entire gray matter were analysed.

With regards to whether the sampling scheme was systematic and evenly spaced across L3–L6:

Our sampling strategy was systematic and based on fixed anatomical levels rather than fixed physical spacing between sections. Specifically, we analysed two spinal cord sections from each lumbar level (L3–L6) per animal, ensuring equivalent representation of each segment of the lumbar enlargement across all genotypes. This is stated in the Immunohistochemistry of spinal cord sections section, where we note that “two sections per lumbar L3–L6 segment were immunostained … per animal.” (lines 198-199). The total number of sections analysed per animal (eight sections: two per lumbar level) is also specified in the Results and figure legends. This anatomical-level–based approach was chosen intentionally because the mouse genotypes used in this study exhibit significant differences in body weight at the same age, which could plausibly result in differences in spinal cord length. Under these conditions, sampling every nth section or at fixed micrometre intervals could lead to unequal representation of lumbar segments across animals. By instead sampling a fixed number of sections from each defined lumbar level, we ensured that the same anatomical regions were analysed consistently across all genotypes. We now clarify this point in the Methods (lines 199-203).

Finally, key biological and classification details should be included—most notably the sex of the mice and whether groups were balanced, given known sex effects in ALS-like phenotypes and spinal cord measures.

Author’s response and action taken:

Please see our response to Reviewer #1’s second comment.

Overall, I recommend major revision. The question is worthwhile and the dataset may be publishable, but the current sample size/power and statistical framing do not yet support the strength of the null conclusions, and additional methodological clarity (and ideally increased N) would substantially improve the rigor and interpretability of the work.

Author’s response and action taken:

We thank the reviewer for their overall assessment and constructive feedback. We have addressed the concerns raised throughout the review by clarifying the methodological details of the sampling and quantification procedures, refining the framing and interpretation of the results, and expanding the discussion of the limitations associated with sample size and statistical power. We believe these revisions improve the clarity and rigor of the manuscript.

---

## [Editor Report · Decision Letter 1]

17 Mar 2026

Impaired dynein function preserves spinal interneuron survival and positioning in an ALS-like mouse model

PONE-D-25-54374R1

Dear Dr. Hafezparast,

We’re pleased to inform you that your manuscript has been judged scientifically suitable for publication and will be formally accepted for publication once it meets all outstanding technical requirements.

Kind regards,

Xiaona Wang, Ph.D

Academic Editor

PLOS One
---

## [Editor Report · Acceptance letter]

PONE-D-25-54374R1

PLOS One

Dear Dr. Hafezparast,

I'm pleased to inform you that your manuscript has been deemed suitable for publication in PLOS One. Congratulations! Your manuscript is now being handed over to our production team.

Kind regards,

on behalf of

Associate Professor Xiaona Wang

Academic Editor

PLOS One